# Sustained Intra-Articular Release and Biocompatibility of Tacrolimus (FK506) Loaded Monospheres Composed of [PDLA-PEG_1000_]-*b*-[PLLA] Multi-Block Copolymers in Healthy Horse Joints

**DOI:** 10.3390/pharmaceutics13091438

**Published:** 2021-09-10

**Authors:** Stefan M. Cokelaere, Wilhelmina M.G.A.C. Groen, Saskia G.M. Plomp, Janny C. de Grauw, Paul M. van Midwoud, Harrie H. Weinans, Chris H.A. van de Lest, Marianna A. Tryfonidou, P. René van Weeren, Nicoline M. Korthagen

**Affiliations:** 1Department of Clinical Sciences, Faculty of Veterinary Medicine, Utrecht University, Yalelaan 112, 3584 CM Utrecht, The Netherlands; cokelaere@sporthorsemdc.com (S.M.C.); w.m.g.a.c.groen@uu.nl (W.M.G.A.C.G.); S.G.M.Plomp@uu.nl (S.G.M.P.); J.C.deGrauw@uu.nl (J.C.d.G.); C.H.A.vandeLest@uu.nl (C.H.A.v.d.L.); M.A.Tryfonidou@uu.nl (M.A.T.); N.M.Korthagen@uu.nl (N.M.K.); 2Sporthorse Medical Diagnostic Centre, Hooge Wijststraat 7, 3584 RC Heesch, The Netherlands; 3Innocore Pharmaceuticals, L.J. Zielstraweg 1, 9713 GX Groningen, The Netherlands; P.vanMidwoud@innocorepharma.com; 4Department of Orthopaedics, University Medical Center Utrecht, Heidelberglaan 100, 3584 CX Utrecht, The Netherlands; h.h.weinans@umcutrecht.nl; 5Department of Biomolecular Health Sciences, Cell Biology and Histology, Faculty of Veterinary Medicine, Utrecht University, Yalelaan 2, 3584 CM Utrecht, The Netherlands

**Keywords:** tacrolimus, prolonged-action preparation, equine, arthritis, biomarkers, synovial fluid

## Abstract

There is an increasing interest in controlled release systems for local therapy in the treatment of human and equine joint diseases, aiming for optimal intra-articular concentrations with no systemic side effects. In this study, the intra-articular tolerability and suitability for local and sustained release of tacrolimus (FK506) from monospheres composed of [PDLA-PEG_1000_]-*b*-PLLA multiblock copolymers were investigated. Unloaded and tacrolimus-loaded (18.4 mg tacrolimus/joint) monospheres were injected into the joints of six healthy horses, with saline and hyaluronic acid (HA) in the contralateral joints as controls. Blood and synovial fluid were analysed for the tacrolimus concentration and biomarkers for inflammation and cartilage metabolism. After an initial burst release, sustained intra-articular tacrolimus concentrations (>20 ng/mL) were observed during the 42 days follow-up. Whole-blood tacrolimus levels were below the detectable level (<0.5 ng/mL). A transient inflammatory reaction was observed for all substances, evidenced by increases of the synovial fluid white blood cell count and total protein. Prostaglandin and glycosaminoglycan release were increased in joints injected with unloaded monospheres, which was mitigated by tacrolimus. Both tacrolimus-loaded monospheres and HA transiently increased the concentration of collagen II cleavage products (C2C). A histologic evaluation of the joints at the endpoint showed no pathological changes in any of the conditions. Together, these results indicate the good biocompatibility of intra-articular applied tacrolimus-loaded monospheres combined with prolonged local drug release while minimising the risk of systemic side effects. Further evaluation in a clinical setting is needed to determine if tacrolimus-loaded monospheres can be beneficial in the treatment of inflammatory joint diseases in humans and animals.

## 1. Introduction

Joint disorders are a major issue for both humans and animals [1,2]. The most prevalent joint disorders in humans are osteoarthritis (OA) and rheumatoid arthritis (RA) [3,4,5,6]. Both conditions are inherently different in background and symptomatology [7], with OA being a degenerative and often localised disorder characterised by low-grade inflammation that may intermittently become more severe and RA being an autoimmune disorder with generally more fulminant inflammation affecting multiple joints [6]. The common factor between these diseases is the chronic nature of the synovial joint inflammation that generally requires lifelong, and ultimately joint replacing, treatment [8,9]. The mainstay of treatment in both diseases is the administration of nonsteroidal anti-inflammatory drugs (NSAIDs) [8]. For both conditions, this treatment is palliative and may alter the clinical symptoms but will not modify the disease. For RA, which is a systemic condition [7], several disease-modifying drugs have been described that can achieve remission to a certain extent [9]. To avoid the well-described adverse effects of long-term systemic treatment with anti-inflammatory or immunosuppressive agents, the local treatment of affected joints is a preferred approach [10]. However, the clearance of most drugs from the joints is very quick, and frequent arthrocentesis is not feasible for reasons of patient welfare and safety [10,11]. Therefore, currently, much effort is being put into the development of controlled or delayed drug release systems for intra-articular application and treatment of joint disease [12,13,14].

A potential drug for these joint disorders is the macrolide antibiotic tacrolimus (FK506, also known as fujimycin). It is an immunosuppressive agent that is used to prevent organ transplant rejection and in the treatment of autoimmune disorders or diseases with an inflammatory component, such as atopic dermatitis, inflammatory bowel disease, polycystic ovary syndrome, and RA [15,16,17,18,19,20,21]. Tacrolimus has multiple modes of action. It suppresses T-cell cytokine production by inhibiting the calcineurin-mediated activation of nuclear factor of activated T cells (NF-AT) [22,23]. Additionally, it functions through NF-AT-independent mechanisms, by the suppression of pathogenic inflammatory cytokine production from monocytes and macrophages [24]. Furthermore, anti-inflammatory effects are exerted through the direct inhibition of chemokine production by rheumatoid synovial fibroblasts [25]. Treatment with tacrolimus can benefit OA and RA patients by providing rapid pain relief in arthritic joints, inhibiting bone/cartilage destruction (and, thus, joint damage progression), and promoting osteogenic and chondrogenic differentiation that may produce favourable effects for bone and cartilage repair [20,21,23,26]. However, oral treatment with tacrolimus is problematic due to toxic effects, especially neurotoxicity and nephrotoxicity [27,28]. Further, tacrolimus has a very narrow therapeutic window and markedly variable oral bioavailability and pharmacokinetics in humans [29], making it difficult to reach the therapeutic levels in affected joints. Tacrolimus is therefore a very suitable candidate for the application of intra-articular controlled delivery.

We previously showed the suitability and potential capacity for the controlled delivery of drugs by monospheres (microspheres with a narrow particle size distribution) based on poly(DL-lactide-PEG)-*b*-poly(L-lactide) multiblock copolymers ([PDLA-PEG_1000_]-*b*-[PLLA]) [30]. Varying the ratio between the relatively hydrophilic, amorphous PDLA-PEG and the rigid, semi-crystalline PLLA allowed the fine-tuning of the drug release kinetics. Another advantage of these copolymers is that drug release occurs through diffusion rather than rapid degradation, preventing the accumulation of degradation products that could affect the bioactivity of the drug [31]. Characterisation, degradation, intra-articular biocompatibility, and drug release of tacrolimus-loaded [PDLA-PEG_1000_]-*b*-[PLLA] based monospheres has been investigated previously *in vitro* and *in vivo*, mainly in rats [32]. It was shown that a prolonged release for up to 42 days was possible *in vitro* [32]. Preliminary studies in healthy horses showed that the local release of tacrolimus into the synovial fluid was detectable for the entire 4-week study period. Some transient inflammation was observed for unloaded monospheres, indicated by elevated levels of white blood cells (WBC) and the total protein (TP) [30,32]; however these appeared to be mitigated by the tacrolimus from the loaded monospheres [32].

Continuing from these preliminary *in vitro* and *in vivo* data, we hypothesised that the intra-articular sustained release of tacrolimus could be achieved for up to 42 days and that the transient inflammatory response to the monospheres was a nonspecific response to the intra-articular injections. In this study, we therefore performed a comprehensive investigation entailing a longer duration of follow-up with a larger sample size and proper controls to identify the potential of [PDLA-PEG_1000_]-*b*-[PLLA] monospheres for the intra-articular drug delivery of tacrolimus in healthy horse joints. The horse has been shown to be a very suitable model for the study of intra-articular drug delivery systems [33,34]. Apart from the obvious practical advantages of joint sizes and easy arthrocentesis, the horse is an animal in which joint disorders are among the most prevalent health issues. Therefore, the horse is a target animal in itself and is seen as one of the best animal models for translational orthopaedic research [35]. To address the objectives, we studied the intra-articular biocompatibility and *in vivo* release kinetics of Tacrolimus-loaded monospheres for 42 days. We compared the responses with the response to a hyaluronic acid (HA) gel that has been registered for clinical application to treat lameness due to non-septic joint disease in the horse. This HA gel is known to elicit a transient inflammatory response and, therefore, serves as a positive control. Besides the analysis of synovial fluid biomarkers for the inflammatory and metabolic status of the joints, we conducted a histological evaluation of the joint tissues.

## 2. Materials and Methods

### 2.1. Preparation of Tacrolimus-Loaded Monospheres

Multiblock [PDLA-PEG_1000_]-*b*-[PLLA] copolymers were synthesised by InnoCore Pharmaceuticals (Groningen, The Netherlands), as described previously [30,32]. L-lactide and DL-Lactide were vacuum-dried for 17 h at 50 °C, and poly(ethylene glycol) with a molecular weight of 1000 g/mol (PEG_1000_) was vacuum-dried for 17 h at 90 °C. 1,4-Butanediol and 1.4 butanediisocyanate were distilled under reduced pressure. Low molecular weight poly(L-lactide) ([PLLA], Mw 4000 g/mol) and poly(DL-lactide)-polyethyleneglycol_1000_-poly(DL-lactide) ([PDLA-PEG_1000_], Mw 2000 g/mol) prepolymers were synthesised by standard stannous octoate catalysed ring-opening polymerisation. To achieve a target molecular weight of 4000 g/mol PLLA, 244.37 g (1.695 mol) L-lactide was added to a three-necked bottle under nitrogen atmosphere. 1,4-Butanediol (5.63 g (62.47 mmol)) was added to commence ring-opening polymerisation, and next, stannous octoate was added at a ratio of 11,500 mol/mol monomer/catalyst. The content was stirred magnetically at 140 °C for 65 h and, afterwards, cooled down to room temperature (RT). The PDLA-PEG_1000_-PDLA prepolymer with a target molecular weight of 2000 g/mol was synthesised by the same procedure using 125 g (0.867 mol) DL-lactide, 125 g (0.125 mol) PEG_1000_ and stannous octoate at a ratio of 13:500 mol/mol monomer/catalyst. PLLA and PDLA-PEG_1000_ prepolymers were then chain-extended to yield 16[PDLA-PEG_1000_]-84[PLLA] multiblock copolymers; PLLA and PDLA-PEG_1000_-PDLA prepolymers were introduced into a three-necked bottle under the nitrogen atmosphere. Next, 65 mL of dry 1,4-dioxane (distilled over sodium wire) was introduced to obtain a 30 wt% prepolymer solution, and the solution was heated to 80 °C to dissolve the prepolymers. Subsequently, 4.23 g (30.18 mmol) of 1,4-butanediisocyanate was added. The reaction mixture was stirred mechanically for 20 h, cooled down to RT and transferred into a tray; after which, it was frozen and vacuum-dried at 30 °C to eliminate 1,4-dioxane.

Microspheres with a target diameter 30 µm (range 28–40 µm) and with a narrow size distribution (referred to as ‘monospheres’) were prepared under the best clean conditions by membrane emulsification-based solvent extraction/evaporation. This selection was based on the findings of our previous studies, where 30 µm was shown to be the most suitable size for intra-articular delivery due to its retention for several weeks and only limited phagocytosis [30,32]. Approximately 0.5 g of 16[PDLA-PEG_1000_-PDLA]-84[PLLA] was dissolved in 1.5 mL dichloromethane (DCM, p.a. stabilised with EtOH, Across, Geel, Belgium) to obtain a 20% *w/w* solution which was subsequently filtered through a 0.2 mm PTFE filter. In the case of Tacrolimus (FK506)-loaded monospheres, 450 mg 16[PDLA-PEG_1000_-PDLA]-84[PLLA] was co-dissolved with 50 mg of tacrolimus (LC laboratories, Woburn, MA, USA) in 1.5 mL of DCM to obtain a 20% *w/w* polymer solution. The filtered polymer solution (DP) was processed through a microsieve membrane (Nanomi BV, Oldenzaal, The Netherlands) at an approximate rate of 0.12 mL/min into an aqueous solution containing 4% *w/v* PVA (CP). The CP/DP volume ratio was around 35 *v/v*. The formed emulsion was stirred over a period of 3 h at RT to extract and evaporate the DCM. Hardened monospheres were collected by centrifugation at 2000 rpm for 3 min, washed twice with demi water and twice with the 0.05% *w/v* aqueous Tween-20 (Across, Geel, Belgium) solution, and lyophilised. Characteristics of the monospheres are represented in Table 1.

The monospheres were reconstituted in saline (B. Braun, Melsungen, Germany) for injection into horses through a 21G needle, which is the size mostly used for arthrocentesis in the horse.

### 2.2. In Vivo Experimental Set Up

The study design was approved by the institutional Ethics Committee on the Care and Use of Experimental Animals in compliance with the Dutch legislation on animal experimentation (number 2013.III.09.063). Two hundred milligrams of unloaded monospheres dispersed in 3 mL saline were administered into the right middle carpal joint of 6 healthy adult warmblood horses (mean ± s.d. age 5.5 ± 2.3 years, bodyweight 470 ± 35 kg, 3 males and 3 females), with clinically and radiographically normal carpal and talocrural joints. Three millilitres of saline were injected into the contralateral left middle carpal joint as a negative control. Two millilitres of a registered HA gel (Hyonate^®^, Bayer Animal Health, Leverkusen, Germany) was injected into the left talocrural joint as a positive control. Two hundred milligrams of tacrolimus-loaded monospheres (9.2% loading; 18.4 mg tacrolimus) in 3-mL saline were injected into the right talocrural joint. For all the injections, a 21G needle was used. The study design is illustrated in Figure 1A.

### 2.3. Evaluation of Clinical Response to the Treatment

Lameness was semi-quantitatively evaluated by an experienced clinician using the 0 to 5 scale as established by the American Association of Equine Practitioners [36]. In this scale, (0) means no lameness, (1) is a lameness that is inconsistently apparent under special circumstances (such as on the incline or on a hard surface); (2) is a subtle lameness that is consistently apparent under special circumstances; and (3–5) are cases of obvious lameness that are consistently present at trotting (3), walking (4), or during the stance (5). Lameness examinations were conducted at timepoint 0, after 8 h, 24 h, 72 h, and then every week (Figure 1B) post-injection. Horses were monitored throughout the study for signs of discomfort.

### 2.4. Collection of Synovial Fluid and Plasma

Synovial fluid samples (2 mL) from the treated joints were aspirated at the same timepoints as the lameness evaluations (Figure 1B). A portion of the synovial fluid was placed in EDTA tubes (BD, Franklin Lakes, NJ, USA) for the WBC count and TP measurements. The remainder was centrifuged in plain tubes at 5000× *g* for 10 min at RT, aliquoted and stored at −80 °C until determination of the tacrolimus content and biomarker analysis. Blood was collected according to the same time scheme from the left jugular vein of the horses in heparinised vials and spun down for 5 min at 1500× *g* to produce plasma. In the plasma samples, the tacrolimus concentration was determined to assess the systemic levels that might result from local applications.

### 2.5. Tacrolimus Concentrations in the Synovial Fluid and Serum

To determine the tacrolimus concentrations in whole blood and synovial fluid samples, tacrolimus was extracted as described earlier [32]. Blood or hyaluronidase-treated synovial fluid samples (100 μL) were transferred into a 1.5 mL test tube, and 200 μL precipitation reagent (methanol/1.125-M ZnSO_4_ in water (66/34, *v/v*) containing 20 ng/mL sirolimus (Sigma-Aldrich, Zwijndrecht, The Netherlands) as the internal standard) was added. Samples were subsequently vortexed for 30 s and left 5 min at RT. After being vortexed for an additional 5 s, the tubes were centrifuged for 10 min at 15,000× *g* at 4 °C. The supernatant was transferred into an autosampler vial, and a 5-μL sample was injected onto a HyPURITY C18 (50 × 2.1 mm, particle size of 3 μm) analytical column (Thermo Fisher Scientific, Utrecht, The Netherlands). Separation was performed at a flow rate of 500 μL/min with a total run time of 3 min. The mobile phases consisted of 10 mM ammonium acetate pH 3.5 in water (A) and 10 mM ammonium acetate pH 3.5 in methanol (B). The following gradient was applied to the column: *A*/*B v/v*: 0–0.8 min, 65/35; 0.8–0.9 min, 21/79; 0.9–2.0 min, 21/79 to 13/87; 2.0–2.1 min, 13/87 to 0/100; 2.1–2.6 min, 0/100; 2.6–2.7 min, 0/100 to 65/35; 2.7–3.2 min 65/35 at a column temperature of 40 °C. The first 0.8 min of the column effluent was discarded to prevent nonvolatile components from entering the ionisation interface, whereafter, the effluent was introduced via an electospray ionisation (EPI) interface (Sciex, Toronto, ON, Canada) into a 4000 Q TRAP mass spectrometer. For maximal sensitivity and for the linearity of the response, the mass spectrometer was operated in multiple-reaction monitoring (MRM) mode at units of mass resolution. Peaks were identified by comparison of the retention time and mass spectra of the standards. For each component, one ion transition was monitored, sirolimus: 931.6 → 864.4 (collision energy: 23 V), and tacrolimus: 821.5 → 768.4 (collision energy: 26 V). The following MS parameters were used: curtain gas: 10 psi, ion spay voltage: 5500 V, source temperature: 360 °C, gas flow 1: 50 psi, gas flow 2: 40 psi, decluster potential: 80 V, and entrance potential: 10 V. Data were analysed with Analyst software version 1.6.2 (Applied Biosystems, Nieuwerkerk a/d IJssel, The Netherlands). Tacrolimus peak areas were corrected for the sirolimus recovery, and concentrations were calculated using a tacrolimus reference line ranging from 0.5 ng to 1000 ng/mL that was linear in this range (*r* = 0.999).

### 2.6. Synovial Fluid Analysis

The synovial fluid WBC and TP concentrations were determined using a Coulter Counter^®^ Z1 (Beckman Coulter, Inc., Brea, CA, USA) and refractometer (Van der Waal instruments, Kamperveen, The Netherlands), respectively [36,37], following the clinical practice. Synovial fluid samples were also evaluated for glycosaminoglycan (GAG) concentrations as a marker for proteoglycan release using a modified 1,9-dimethylmethylene blue dye-binding assay, as previously described [38]. To check for possible damage to the collagen network of the cartilage, we also measured the concentration of C2C, a neo-epitope present on collagenase-cleavage fragments of type II collagen, employing a commercial ELISA kit (IBEX Technologies, QC, Canada) in accordance with the manufacturer’s recommendations. All assays have previously been validated for use in horses [38,39].

The prostaglandin E_2_ (PGE_2_) synovial fluid concentrations were determined by a high-performance liquid chromatography (HPLC)–tandem mass spectrometry (MS/MS) analysis, as described previously [40]. Briefly, the samples were recovered in a total volume of 1 mL of 15% (*v/v*) methanol + 0.5% glacial acetic acid in the presence of 10 pg/µL 16,16-dimethyl PGF_2α_(Cayman Chemical Company, Ann Arbor, MI, USA) that served as an internal standard. The samples were separated on C18 SPE columns (Merck, Darmstad, Germany). The eicosanoids were eluted with 2 × 0.35 mL ethyl acetate and evaporated to dryness under nitrogen. Evaporated samples were reconstituted in 50 μL of 50% ethanol and subject to a HPLC–MS analysis. Multiple reaction monitoring (MRM) was used as described previously [40]. The supernatant was transferred into an autosampler vial, and a 10 μL sample was injected onto a Luna C18 (2.5 μm 100 × 3 mm; Phenomenex, Torrance, CA, USA). Separation was performed at a flow rate of 200 μL/min with a total run time of 25 min. The mobile phases consisted of 0.02% glacial acetic acid in water (A) and 0.02% glacial acetic acid in acetonitrile (B). The following gradient was applied to the column; A/B *v/v*: 0–1 min, 80/20; 1–17 min, 63/37 to 52/48; 17–18 min, 52/48 to 13/87; 18–23 min, 0/100 and 24 to 25 min, 80/20 at ambient temperature. The effluent was introduced to an EPI interface into a 4000 QTRAP mass spectrometer. For maximal sensitivity and for linearity of the response, the mass spectrometer was operated in multiple-reaction monitoring (MRM) mode at unit mass resolution. Peaks were identified by the comparison of retention time and mass spectra of standards. For each component, one ion transition was monitored, PGE_2_: 351.2 → 271.2 (collision energy: −25 V) and 16,16-dimethyl PGF_2α_ (IS): 381.2 → 319.2 (collision energy: −35 V). The following MS parameters were used: curtain gas: 10 psi, ion spay voltage: −4500, source temperature: 350 °C, gas flow 1: 50 psi, gas flow 2: 40 psi, decluster potential: −100 V, and entrance potential: −10 V. Data were analysed with Analyst software version 1.6.2 (Applied Biosystems, Nieuwerkerk a/d IJssel, The Netherlands). PGE_2_ peak areas were corrected for IS recovery, and concentrations were calculated using a PGE_2_ reference line ranging from 10 to 1000 pg on the column that was linear in this range (r = 0.99).

The PGE_2_ results from 8 h had to be excluded because of too many missing samples for that timepoint, due to insufficient amounts of synovial fluid. Furthermore, at 1, 3 and 5 weeks, the C2C synovial fluid levels were not determined because of the limited synovial fluid volume availability.

### 2.7. Histological Analysis and Grading of Articular Cartilage and Synovial Lining

Six weeks after the start of the study, the animals were euthanised. Directly after, macroscopic scoring of the injected joints was performed according to McIlwraith et al. [41]. This system scores gross changes of the cartilage surface. More specifically, wear lines, erosions, and palmar arthrosis were scored 0–3, where 0 = no changes, 1 = 1 to 2 partial thickness lines or <5 mm diameter erosions, 2 = 2 to 5 partial or 1 to 2 full thickness lines or >5 mm diameter erosions, and 3 = > 5 partial or >2 full thickness lines or full thickness erosions. Next, the samples were harvested and processed for histology as follows; in the middle carpal joints, opposing articular weightbearing surfaces (i.e., third and radial carpal bone articular surface) were harvested; in the talocrural joints, the articular surface of the medial talar ridge was harvested. These osteochondral explants were decalcified in 0.5 M EDTA (Sigma-Aldrich, Zwijndrecht, The Netherlands). Furthermore, from each joint, approximately 10 mm^2^ of synovial membrane was harvested randomly throughout the joint. All samples were fixed in buffered formaldehyde 4% solution (Klinipath, Duiven, The Netherlands) and embedded in paraffin (Sigma-Aldrich, Zwijndrecht, The Netherlands). Sections 5 µm thick were deparaffinised, and the cartilage samples were stained with Safranin O/Fast Green to stain glycosaminoglycans red and collagen green, while synovial membrane samples were stained with eosin, and the cell nuclei were counterstained with haematoxylin, as previously described by Gawlitta et al. [42]. All sections were then mounted in DPX (Millipore, Burlington, MA, USA), and micrographs were taken with an optical microscope (Olympus BX51, Olympus, Germany). A modified Mankin scoring system for histopathological grading was performed on osteochondral samples and a microscopic grading system on synovial membrane sections as described in McIlwraith et al. (Tables 5 and 6, respectively) [41]. All scorings were performed by two independent researchers (SC and SP) in a blinded fashion.

### 2.8. Statistical Analysis

Tacrolimus concentrations are presented as the mean ± SD. Synovial fluid data (WBC, TP, GAG content, C2C, and PGE_2_ content) are presented as the mean ± SD and were compared using generalised linear mixed models for repeated measures with “treatment” as the fixed effect and “donor” as the random effect and a log link for gamma distributions. To correct for multiple testing, the Benjamini–Hochberg (false discovery rate) procedure was performed; *p*-values were ranked, and for each *p*-value, a critical value was calculated by the formula (i/m)Q, where i = individual rank, m = number of tests, and Q = false discovery rate (0.25). The largest *p*-value smaller than the corresponding critical value was used as a cut-off value for significance. Adjusted *p*-values were calculated as *p*(i/m). Normality of the histological scores was tested using a Shapiro–Wilk test; for cartilage scores, the data was log-transformed. The scores were compared using a two-way ANOVA and analysed using the inter-observer agreement and a Bland–Altman plot to detect possible bias. Statistical analyses were performed using computer software (SPSS 25 for Windows, IBM, Armonk, NY, USA and GraphPad Prism 7 for Mac, GraphPad Software, San Diego, CA, USA). The level of significance was set at *p* ≤ 0.05.

## 3. Results

### 3.1. Clinical Response to the Intra-Articular Injections

No changes in appetite, pulse, or respiration were observed, and the rectal temperature remained within the normal limits throughout the entire experimental period. All the horses were free of lameness prior to injections, and none of the limbs injected with unloaded monospheres, and tacrolimus-loaded monospheres showed signs of lameness throughout the study. However, one horse showed signs of severe joint distention and subtle lameness in the joint injected with HA (lameness score on a 5-point scale was 1 to 2 at 8 h and 24 h post-injection). The lameness was only temporary and had disappeared at 72 h post-injection. The joint distention of the affected joint in this horse decreased to a moderate degree and remained in this state throughout the whole study. All other horses showed a mild transient joint distention from 8 h to 72 h post-injection in the joint injected with HA. Furthermore, one horse showed a very mild irregular lameness classified as 0.5–1 out of 5 at 72 h post-injection in the joint injected with saline. 

### 3.2. Release Kinetics of Tacrolimus-Loaded Monospheres in Healthy Horse Joints

A tacrolimus concentration was measured in the synovial fluid of all six horses after injection of the loaded monospheres (Figure 2). The tacrolimus synovial fluid levels remained elevated during the entire follow-up period of 42 days. The C_max_ of tacrolimus in synovial fluid was 1848 ± 470 ng/mL and was observed at 8 h post-injection. After reaching C_max_, the tacrolimus concentration in the joint decreased rapidly during the first week, to decrease much more slowly in the following period. The concentration was 80.0 ± 16.5 ng/mL and 20.0 ± 5.8 ng/mL at 7 and 42 days post-injection, respectively. The total area under the curve (AUC) was 3137 ± 838 ng × h/mL. Whole-blood tacrolimus levels were below the detectable drug concentration (<0.5 ng/mL) at all timepoints.

### 3.3. Local Inflammatory Response

Figure 3 shows the local inflammatory response of the joints prior to and after the injection of saline, unloaded monospheres, HA, and tacrolimus-loaded monospheres. The synovial fluid of all injected joints showed a transient increase of the WBC count with a peak 8 h post-injection (>50 × 10^9^ cells/L; Figure 3A). At 24 h post-injection, the WBC count in joints treated with saline, unloaded monospheres, and tacrolimus-loaded monospheres remained significantly increased (*p* < 0.0001, *p* = 0.0005, and *p* = 0.0042, respectively), while, in the joints treated with HA, it almost returned to the baseline values. At 72 h post-injection, the WBC count of all the joints returned to below the baseline values and remained so until the end of the study.

At 8 h post-injection, the synovial fluid from joints injected with unloaded monospheres significantly increased the TP concentrations compared to the synovial fluid of joints injected with saline, HA, and tacrolimus-loaded monospheres (*p* = 0.0066, *p* = 0.0076, and *p* = 0.0086, respectively; Figure 3B). At 24 h post-injection, TP in the synovial fluid from joints injected with unloaded monospheres, HA, and tacrolimus-loaded monospheres increased and became significantly higher than in the saline control for all groups (*p* < 0.0001). At 72 h post-injection, the TP concentration in all the injected joints started to decrease again and was no longer significantly increased at week 1 post-injection for saline and at week 2 for tacrolimus-loaded monospheres compared to the baseline values. For HA controls, it took 5 weeks to return to the baseline values, where, for unloaded monospheres, the TP values did not return to baseline but were not significantly higher anymore at week 6 post-injection. Furthermore, in joints injected with unloaded monospheres, the TP remained significantly higher than in joints injected with saline up to week 2, while, for tacrolimus-loaded monospheres, the TP values were significantly lower than saline controls from week 3 until week 6. From week 1 on, almost all the values were below the clinical cut-off value of 2.5 g/dL, except for one donor, who revealed an increased TP in weeks 4 and 5 in all joints, except for the joint injected with tacrolimus-loaded monospheres.

PGE_2_ concentrations (Figure 3C) showed a transient increase for unloaded monospheres (*p* < 0.0001) and HA (*p* = 0.0004) at 24 h. Additionally, there was a significantly lower level for tacrolimus-loaded monospheres compared to unloaded monospheres at 24 h (*p* = 0.0004).

### 3.4. Response of Synovial Fluid Markers of Cartilage Metabolism

To determine whether high tacrolimus concentrations or the transient inflammatory response had any detrimental effect on the cartilage, the GAG content and concentration of the C2C epitope of collagen in the synovial fluid were measured (Figure 4A,B). GAG levels were significantly increased at 24 h and/or 72 h post-injection compared to the baseline for all joints (*p* < 0.0001; Figure 4A). Additionally, the GAG levels were significantly higher in joints injected with HA than in joints injected with unloaded and tacrolimus-loaded monospheres (*p* = 0.0043 and *p* < 0.0001, respectively, at 24 h). After 72 h, the GAG levels in joints injected with unloaded monospheres were still significantly elevated compared to saline (*p* = 0.0031) and tacrolimus-loaded monospheres (*p* = 0.0006). The GAG concentrations were completely back to baseline after 1 week for all treatments. The concentration of C2C, a biomarker for collagen cleavage, increased significantly compared to the baseline. For unloaded monospheres, HA, and tacrolimus-loaded monospheres, this reached significance after 24 h and, for saline, after 72 h (*p* = 0.0230, *p* = 0.0038, *p* = 0.0005, and *p* < 0.0001, respectively; Figure 4B).

### 3.5. Histological Analysis

A histological examination of the synovial membrane of the joints from this study did not show signs of granulomatous synovitis or any other indication of substantial inflammation (Figure 5). There were no significant differences in the total synovial membrane histological grading score between saline, unloaded monospheres, HA, and tacrolimus-loaded monospheres (Table 2). In all six horses, samples taken from the synovium at the 6-week endpoint revealed the presence of monospheres in both joints injected with unloaded and tacrolimus-loaded monospheres.

Macroscopically, the cartilage surface was smooth and had a normal appearance in all the injected joints. All joints received a score “0”. At the histological level, there were no significant differences in the cartilage between saline, unloaded monospheres, HA, and tacrolimus-loaded monospheres (4.8 ± 1.2, 6.3 ± 1.4, 7.7 ± 2.4 and 6.0 ± 1.4, respectively; Figure 5).

## 4. Discussion

We have previously shown promising results for tacrolimus-loaded monospheres in a small pilot study with three horses [32]. The current larger study was intended to confirm and expand these results by studying the *in vivo* application of tacrolimus-loaded poly(DL-lactide-PEG_1000_)-*b*-poly(L-lactide)-based monospheres more in-depth in healthy horse joints. 

### 4.1. Release Kinetics from Tacrolimus-Loaded Monospheres in Healthy Horse Joints

The release kinetics of tacrolimus from the [PDLA-PEG_1000_]-*b*-[PLLA] monospheres are similar to those reported previously [32] and are likely governed by a combination of diffusional and polymer degradation mechanisms. The tacrolimus concentration in the synovial fluid also depends on the elimination rate of the drug from the joint, which is hypothesised to be very rapid due to the large concentration difference between the joint and the bloodstream and the relatively small molecular size of the drug. Additionally, clearance rates may depend on the metabolisation rate, which can vary considerably between individuals [43], and on the disease state/condition of the joint [44,45]. Compared to the *in vitro* results, a more rapid release, together with the absence of a lag phase, was observed *in vivo* [32]. This indicates that the environment in the joint favours an initial release of tacrolimus from the monospheres.

Tacrolimus has only been used in the horse in the form of a topically applied ointment to treat hyperkeratosis [46] and specific eye pathology [47], so little is known about the effective concentration of tacrolimus, either systemically or locally. In humans, the drug is known to have a narrow therapeutic index of 3–20 ng/mL measured in whole blood. While tacrolimus is predominantly metabolised by the liver, the upper limit of the systemic therapeutic index is mainly dictated by nephro-, hepato-, and neurotoxicity [27,28], with maximum systemic levels of 10–20 ng/mL, depending on the indication [29,48,49]. In this study, the systemic drug concentrations were below the detection limits (<0.5 ng/mL), even at 8 h post-injection (C_max_). This confirms the feasibility of site-specific drug delivery in the joint via intra-articular injection without resulting in elevated systemic levels. Additionally, no systemic adverse reactions were seen; however, a complete pathological examination of the animals was not performed. Although it is possible that the systemic levels were above the detection limit at time points that were not sampled in this study, the detection limit of 0.5 ng/mL was 10 times lower than the clinical target level in humans, so systemic side effects are less likely to occur with the intra-articular delivery of tacrolimus.

In the current study, the average tacrolimus concentrations measured in synovial fluid were >20 ng/mL throughout the entire study duration of 42 days, so the intra-articular concentrations were well above the optimal therapeutic range in whole blood. C_max_ in the synovial fluid was nearly 2000 ng/mL, which is 100-fold the desired systemic concentration in humans. High local concentrations are, however, not harmful per se; in our previous study, doses of up to 1000 ng/mL did not affect the proliferation and viability of human articular chondrocytes *in vitro* [30]. Additionally, in humans, concentrations of 20–40 mg/mL are being applied rectally, gaining peak levels of 250 ng/mg in human colonic mucosal tissue [16].

Based on the findings of the current and previous studies, it can be concluded that the use of poly(DL-lactide-PEG)-*b*-poly(L-lactide)-based monospheres loaded with tacrolimus greatly increases the tacrolimus concentrations in synovial fluid while drastically reducing the exposure to systemic concentrations of the drug and, hence, the potential occurrence of systemic side effects.

### 4.2. Response to the Intra-Articular Injections

#### 4.2.1. Inflammatory Response

No systemic side effects were noticed throughout the whole study period. Furthermore, there were no clinical signs of lameness originating from the joints injected with monospheres. From this, it can be concluded that the intra-articular application of monospheres has no negative clinical effect on locomotion and is tolerated well. After intra-articular injection of the unloaded and tacrolimus-loaded monospheres in the joint, a transient inflammatory reaction occurred, indicated by increased WBC and TP in the synovial fluid. This confirms what was reported in the previous study [32]. Likewise, PGE_2_ concentrations in the synovial fluid transiently increased at 24 h post-injection of unloaded monospheres, whereas the WBC was similar in both joints injected with loaded and unloaded monospheres, the TP and PGE_2_ levels were higher in joints injected with unloaded monospheres than in joints injected with tacrolimus-loaded monospheres. Given that PGE_2_ is a known vasodilator [50], together with the fact that PGE_2_ has immune-regulating characteristics [51], it is fair to reason that unloaded monospheres trigger an immune response to some extent. This immune response was most likely reduced by tacrolimus from the loaded monospheres, confirming the drug’s immunosuppressive effect. As such, the most straightforward conclusion is that the anti-inflammatory effect of tacrolimus mitigated the inflammatory response to the monospheres.

A transient increase in WBC and TP was also present after the intra-articular injection of the control substances saline and HA. For HA, this was accompanied by a clinically significant joint distention in all the horses. One horse had a severe distention, which is known as a “flare”. This temporary adverse clinical symptom is the consequence of an acute inflammatory reaction and is not uncommon in horses after an intra-articular injection with HA-containing pharmaceutical products [52]. These reactions may induce very high WBC and a substantial rise in TP to an extent that is mostly associated with septic arthritis but are of a rapidly transient nature. Similar responses have been reported after the injection of (sterile) lipopolysaccharide (LPS) and, also, to a lesser extent, as a transient reaction to the injection of saline. They are likely a more general response of some individuals to the arthrocentesis itself and the injection of (any) material into the joint [32,52,53,54]. For HA, this inflammatory response can be quite severe and was, in this case, still detectable after 6 weeks, as shown by the mild increase in synovial grading score. These results once again indicate the importance of using relevant controls when evaluating biocompatibility in healthy joints.

All inflammatory effects in the synovial fluid that were triggered by the monospheres had long since subsided at 6 weeks post-injection, which was proven by a histopathological evaluation of the synovial membrane (Figure 5). The remaining, visible presence of monospheres in the synovial membrane coincided with previous findings of a presence of monospheres in the middle carpal joints of horses at 4 weeks and at 90 days in rat knees [32]. Such a presence could not be other than expected, given the ongoing release of tacrolimus in the synovial fluid at 6 weeks. These findings could have implications for the interval of repeated therapeutic applications, as monospheres and their degradation by-products could accumulate in the joint after multiple injections. Besides identifying the optimal therapeutic dose and interval for the intra-articular application of tacrolimus, more research is needed to evaluate the *in vivo* degradation rates of the [PDLA-PEG_1000_]-*b*-[PLLA] monospheres and the possible systemic effects of the degradation products thereof on the joint tissues.

#### 4.2.2. Local Effect on Cartilage

To determine whether any of the intra-articular injections had a detrimental effect on the cartilage, markers for cartilage matrix metabolism were measured in the synovial fluid samples. A transient glycosaminoglycan release, indicated by elevated GAG levels in synovial fluid, was seen shortly after injection of all substances except for saline. This means that the injection of unloaded monospheres, tacrolimus-loaded monospheres, or HA has an effect on the cartilage indeed, but its nature and clinical relevance remain elusive. High GAG levels in the synovial fluid are known to correlate with the progression of OA in humans [55]. However, in horses, GAG levels in synovial fluid are also known to increase due to exercise [56] and may thus be the result of a stimulation of the metabolism of healthy chondrocytes with a related increase in GAG turnover, as well as of cartilage damage, and can hence be either catabolic or anabolic in nature. If a GAG release in response to unloaded monospheres would be due to catabolic processes, this effect was most likely mitigated by tacrolimus.

The catabolic marker for collagen degradation, C2C, increased slightly for all substances but only significantly for tacrolimus-loaded monospheres and HA. It is unclear what caused these elevated C2C levels. Here, too, it is possible that it reflects a general increase of metabolism of the extracellular matrix components rather than a sign of tissue degradation. Unfortunately, synovial fluid volumes were limited; therefore, anabolic collagen markers or MMPs could not be measured in order to confirm or reject this hypothesis. Similar peaks were observed in our previous study after the injection of LPS (24 h) and saline (72 h) [52]. Overall, the GAG and C2C levels were very similar to those observed in previous studies [32,52]; however, no clear conclusions can be drawn with our current knowledge.

If in the worst case, the minor transient effects that were seen after injection of the materials were merely catabolic, no long-term effect in the sense of the deterioration of tissue was seen at a histopathological examination. Therefore, it seems warranted to state, based on the combination of macroscopic, histological, and synovial fluid biomarker findings, that the injections of the unloaded and tacrolimus-loaded monospheres did not have any obvious deleterious effect on the articular cartilage and were, hence, well-tolerated. This finding is in line with data from previous studies in horses and rats [30,32], and with extensive *in vitro* analyses that showed anticatabolic and even anabolic effects of tacrolimus on chondrocytes [57,58].

## 5. Conclusions

Intra-articular injection of tacrolimus-loaded monospheres in a healthy horse model demonstrated that, after an initial burst release, high local tacrolimus concentrations were achieved in the joint for a prolonged time, while the systemic exposure to tacrolimus was negligible. Furthermore, the histological and synovial fluid analyses showed that the tacrolimus-loaded monospheres were well-tolerated and did not affect the cartilage. Altogether, this study demonstrated that monospheres based on poly(DL-lactide-PEG_1000_)-*b*-poly(L-lactide) multiblock copolymers loaded with tacrolimus have potential for successful controlled intra-articular drug delivery for the treatment of joint diseases.

## Figures and Tables

**Figure 1 pharmaceutics-13-01438-f001:**
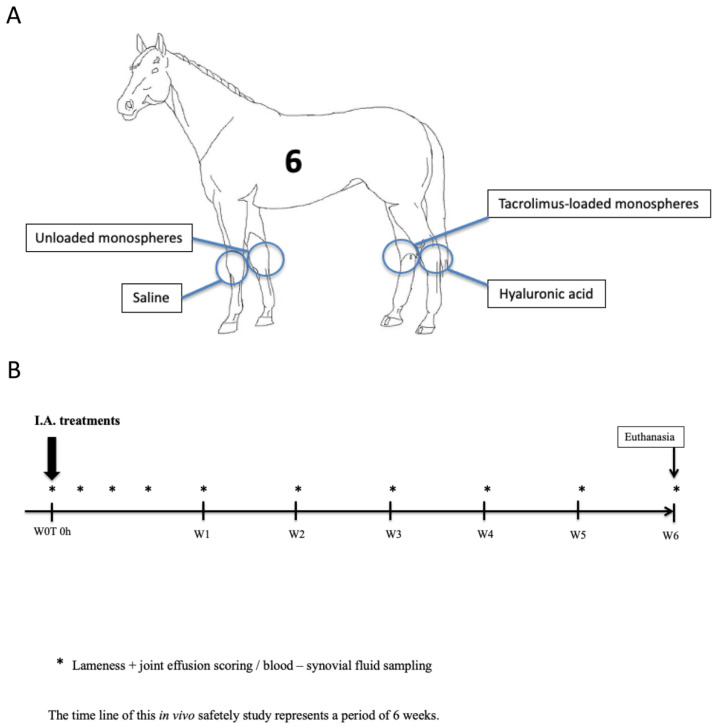
Schematic view of intra-articular treatments in each of the 6 horses (**A**). Unloaded monospheres were administered into the right middle carpal joint and tacrolimus-loaded monospheres were injected into the right talocrural joint. Saline was injected into the left middle carpal joint as a negative control. HA was injected into the left talocrural joint as a positive control. The timeline shows an overview of the 6-week study period (**B**).

**Figure 2 pharmaceutics-13-01438-f002:**
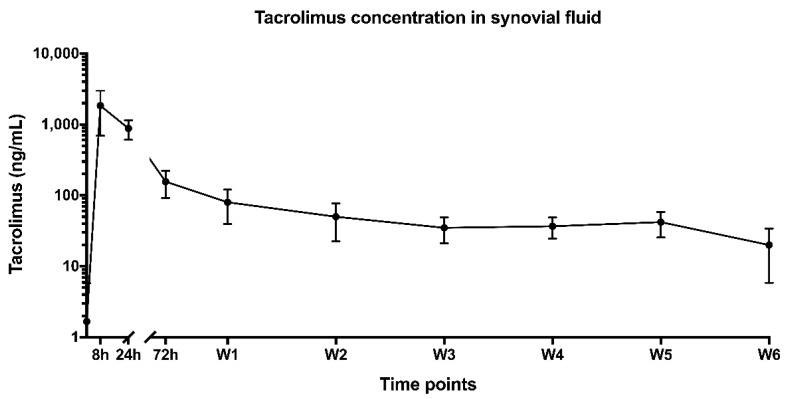
Tacrolimus concentrations in synovial fluid after the intra-articular administration of Tacrolimus loaded in [PDLA-PEG_1000_]-*b*-[PLLA] multiblock copolymers over a 6-week (W1–W6) period. Graph shows the mean ± SD.

**Figure 3 pharmaceutics-13-01438-f003:**
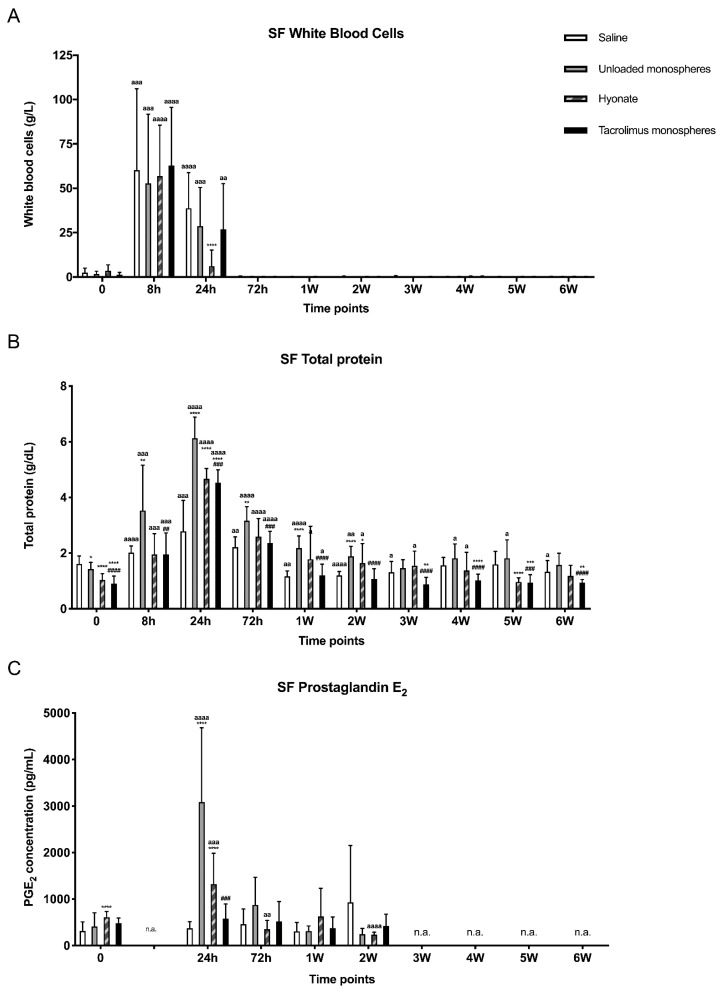
Synovial fluid concentrations of inflammatory parameters: WBC (**A**), TP (**B**), and PGE_2_ (**C**). Graphs show the mean ± SD. Significant differences are depicted as follows: treatment groups vs. saline (*), unloaded vs. tacrolimus-loaded monospheres (#), or differences compared to the baseline within the same treatment group (a). One symbol: *p* > 0.01, two symbols: *p* < 0.01, three symbols: *p* < 0.001, and four symbols: *p* < 0.0001.

**Figure 4 pharmaceutics-13-01438-f004:**
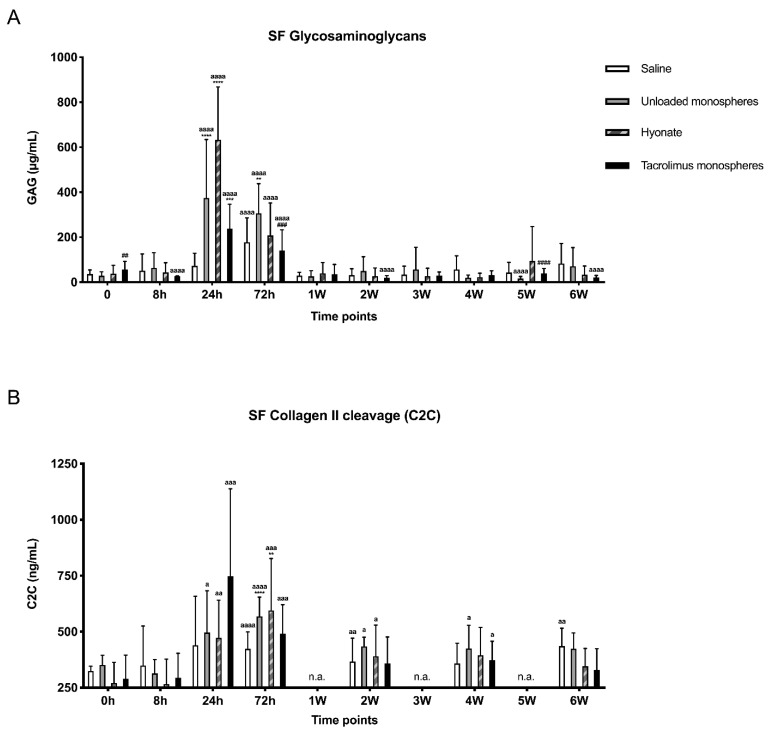
Synovial fluid concentrations of markers for cartilage metabolism over time for all 4 treatment groups: glycosaminoglycans (GAGs) (**A**) and collagen II cleavage marker (C2C) (**B**). Graphs show the mean ± SD. Significant differences are depicted as follows: treatment groups vs. saline (*), unloaded vs. tacrolimus-loaded monospheres (#), or differences compared to the baseline within the same treatment group (a). One symbol: *p* > 0.01, two symbols: *p* < 0.01, three symbols: *p* < 0.001, and four symbols: *p* < 0.0001.

**Figure 5 pharmaceutics-13-01438-f005:**
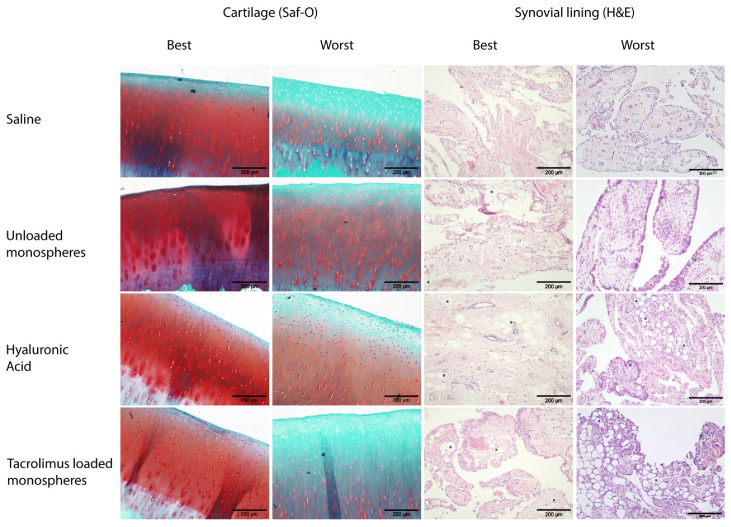
Representative histological images (10× magnification). Cartilage stained with safranin O and synovial tissue stained with H&E. Monospheres in the synovial tissue are indicated with an asterisk (*).

**Table 1 pharmaceutics-13-01438-t001:** Overview of the monospheres with their characteristics.

	Unloaded	Tacrolimus-Loaded
Polymer	16[PDLA-PEG_1000_]-84[PLLA]	16[PDLA-PEG_1000_]-84[PLLA]
Average particle size	37 µm	39 µm
Morphology	Smooth and non-porous ^1^	Smooth and non-porous ^1^
FK506 loading	N.A.	9.2%
Encapsulation efficiency	N.A.	92%
Injection volume per joint	3 mL	3 mL
Monospheres injected per joint	200 mg	200 mg
Dose FK506 injected per joint	0 mg	18.4 mg

^1^ Reprinted with permission from [32]. Copyright 2018 American Chemical Society.

**Table 2 pharmaceutics-13-01438-t002:** Synovial membrane and cartilage microscopic scores according to the scoring system described by Mcllwraith et al. 2010 [41]. Scores are depicted as median (range), and the maximum score is 20 for both scores. No significant differences between treatments were observed.

	Saline	Unloaded Monospheres	Hyaluronic Acid	Tacrolimus Monospheres
Synovial membrane score	5 (4–7)	6.5 (5–8)	8.5 (4–11)	6 (4–8)
Cartilage microscopic grade	4.5 (4–7)	6.5 (4–8)	8 (4–10)	6 (4–8)

## Data Availability

The data presented in this study are available on request from the corresponding author.

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
