# Peer review of "Sustained Intra-Articular Release and Biocompatibility of Tacrolimus (FK506) Loaded Monospheres Composed of [PDLA-PEG1000]-b-[PLLA] Multi-Block Copolymers in Healthy Horse Joints"

_pharmaceutics, 2021, doi:10.3390/pharmaceutics13091438_

Round 1

Reviewer 1 Report

This work represents an in-vivo application of drug delivery agents (polymer microparticles) previosly preparared and characterized by the same research group. The loaded drug is the antibiotic tacrolimus, a potential remedy for joint disorders whose oral administration is problematic due to toxic effects. The rationale for using a carrier is well explained by the authors as a means for suppressing the unwanted side-effects of the drug itself (toxicity and poor bioavailability), while increasing sustained release. In turn, this latter issue is  very important for anti-inflammatory drugs and therefore the main purpose of the present work sounds appropriate in terms of how the subject is presented and how a possible solution is proposed. This study is well conducted and the data obtained are critically analysed/discussed. I have no objection for publishing this manuscript in its current form though I have a minor remark on the name "monoparticle", which is misleading and doesn't provide the reader with immediate significance. It is probably intended as a shorthand for monodisperse microparticle, as it is also explained in the introduction, and moreover it has already been used by the same authors, but not every new acronym is necessarily useful...

Author Response

Dear reviewer, 

Thank you for evaluating our manuscript.

We think you might refer to “monosphere” which represents microspheres with a narrow particle size distribution. This is the term used for the product by its producer Innocore Pharmaceuticals. Therefore, we think it is best to keep using the term “monosphere”, however, if there are severe objections against that term, we are happy to consider other terminology.

We hope the manuscript meets your expectations and that you find it suitable for publication. 

Best regards, 

Lotte Groen

Reviewer 2 Report

The authors have evaluated the sustained intra-articular release and biocompatibility of tacrolimus (FK506) loaded monospheres composed of PDLA-PEG-b-PLLA multi-block copolymers in healthy horse joints. In general, the work has novelty and deserves attention. A systematic study has been carefully done and the results are scientifically interpreted and discussed. However, the following minor issues need to be addressed before this article could be accepted for publication:

  1. “slow-release” or “sustained-release” which will be a better keyword?
  2. Please add the city and country in parenthesis for equipments, chemicals and reagents used in addition to company from where they were bought. In the case of USA, you need to mention the name of state in addition to city and country.
  3. The section 2.1 should be elaborated to include step-by-step procedure for the preparation of tacrolimus loaded monospheres. Even including a flowchart will be better.
  4. A brief description of the HPLC-Tandem MS analysis should be included under section 2.5.
  5. The labels heads on the top of each bar in both Figures 3 and 4 should be increased in size for clarity.
  6. The manuscript should be double-checked for providing the full forms of all the abbreviations in the first instance and abbreviating thereafter.

Author Response

Dear reviewer, 

Thank you for evaluating our manuscript. We have answered your question point by point below. 

  1. Good point, we changed the keyword to prolonged-action preparation which is covered by the MeSH term “Delayed-Action Preparations”. As sustained release is already present in the title we feel a different keyword is adding to the findability of the manuscript. Furthermore, there is an initial burst release, therefore controlled release does not cover the content. We would like to keep the term “sustained release” in the title as it best suits the content.
  2. We added the required information in the material and methods in the manuscript.
  3. We adjusted section 2.1:

    Multi-block [PDLA-PEG1000]-b-[PLLA] co-polymers were synthesized by InnoCore Pharmaceuticals as described previously [30,32]. L-lactide and DL-Lactide were vacuum dried for 17h at 50°C and poly(ethylene glycol) with a molecular weight of 1000 g/mol (PEG1000) was vacuum dried for 17h at 90°C. 1,4-Butanediol and 1.4 butanediisocyanate were distilled under reduced pressure. Low molecular weight poly(L-lactide) ([PLLA], Mw 4000 g/mol) and poly(DL-lactide)-polyethyleneglycol1000-poly(DL-lactide) ([PDLA-PEG1000],Mw 2000 g/mol) prepolymers were synthesized by standard stannous octoate catalysed ring-opening polymerization. To achieve a target molecular weight of 4000 g/mol PLLA, 244.37 g (1.695 mol) L-lactide was added to a three-necked bottle under nitrogen atmosphere. 1,4-Butanediol (5.63 g (62.47 mmol)) was added to commence ring-opening polymerization and next, stannous octoate was added at a ratio of 11500 mol/mol monomer/catalyst. The content was stirred magnetically at 140°C for 65 h and afterwards cooled down to room temperature (RT). PDLA-PEG1000-PDLA prepolymer with a target molecular weight of 2000 g/mol, was synthesized by the same procedure using 125 g (0.867 mol) DL-lactide, 125 g (0.125 mol) PEG1000 and stannous octoate at a ratio of 13500 mol/mol monomer/catalyst. PLLA and PDLA-PEG1000 prepolymers were then chain-extended to yield 16[PDLA-PEG1000]-84[PLLA] multiblock co-polymers; PLLA and PDLA-PEG1000-PDLA pre-polymers were introduced into a three-necked bottle under nitrogen atmosphere. Next, 65 ml dry 1,4-dioxane (distilled over sodium wire) was introduced to obtain a 30 wt% pre-polymer solution and the solution was heated to 80°C to dissolve the prepolymers. Subsequently, 4.23 g (30.18 mmol) of 1,4-butanediisocyanate was added. The reaction mixture was stirred mechanically for 20h, cooled down to RT and transferred into a tray, after which it was frozen and vacuum dried at 30°C to eliminate 1,4-dioxane.

    Microspheres with a target diameter 30 µm (range 28-40 µm) and with a narrow size distribution (referred to as ‘monospheres’) were prepared under best clean conditions by membrane emulsification-based solvent extraction/evaporation. This selection was based on findings of our previous studies where 30 µm showed to be most suitable size for intra-articular delivery due to its retention for several weeks and only limited phagocytosis [30,32]. Approximately 0.5 g of 16[PDLA-PEG1000-PDLA]-84[PLLA] was dissolved in 1.5 mL dichloromethane (DCM, p.a. stabilized with EtOH, Across, Geel, Belgium) to obtain a 20% w/w solution which was subsequently filtered through a 0.2 mm PTFE filter. In case of Tacrolimus (FK506)-loaded monospheres, 450 mg 16[PDLA-PEG1000-PDLA]-84[PLLA] was co-dissolved with 50 mg of tacrolimus (LC laboratories, Woburn, Massachusetts, USA) in 1.5 mL of DCM to obtain a 20% w/w polymer solution. The filtered polymer solution (DP) was processed through the microsieve membrane (Nanomi BV, Oldenzaal, The Netherlands) at an approximate rate of 0.12 mL/min into an aqueous solution containing 4% w/v PVA (CP). The CP/DP volume ratio was around 35 v/v. The formed emulsion was stirred over a period of 3h at RT to extract and evaporate DCM. Hardened monospheres were collected by centrifugation at 2000 rpm for 3 min, washed twice with demi water and twice with 0.05% w/v aqueous Tween-20 (Across, Geel, Belgium) solution and lyophilized. Characteristics of the monospheres are represented in table 1.

    The monospheres were reconstituted in saline (B. Braun, Melsungen, Germany) for injection in horses through a 21G needle, which is the size mostly used for arthrocentesis in the horse.

  4. We adjusted section 2.5, adding detailed descriptions of the MS analyses:

    2.5. Tacrolimus concentrations in the synovial fluid and serum

    To determine the tacrolimus concentrations in whole blood and synovial fluid samples, tacrolimus was extracted as described earlier [32]. Blood or hyaluronidase treated synovial fluid samples (100 μl) were transferred into a 1.5 ml test tube, and 200 μl precipitation reagent (methanol/1.125 M ZnSO4 in water (66/34, v/v) containing 20 ng/ml sirolimus (Sigma-Aldrich, Zwijndrecht, The Netherlands) as internal standard) was added. Samples were subsequently vortexed for 30 s and left 5 min at room temperature. After being vortexed for an additional 5 s, the tubes were centrifuged for 10 min at 15 000 g at 4 °C. The supernatant was transferred into an autosampler vial, and a 5 μL sample was injected onto a HyPURITY C18 (50 × 2.1 mm, particle size of 3 μm) analytical column (Thermo Fisher Scientific, Utrecht, NL). Separation was performed at a flow rate of 500 μl/min with a total run time of 3 min. The mobile phases consisted of 10 mM ammonium acetate pH 3.5 in water (A) and 10 mM ammonium acetate pH 3.5 in methanol (B). The following gradient was applied to the column; A/B vol/vol: 0–0.8 min, 65/35; 0.8–0.9 min, 21/79; 0.9–2.0 min, 21/79 to 13/87; 2.0–2.1 min, 13/87 to 0/100; 2.1–2.6 min, 0/100; 2.6–2.7 min, 0/100 to 65/35; 2.7–3.2 min, 65/35 at a column temperature of 40 °C. The first 0.8 min of the column effluent was discarded to prevent nonvolatile components to enter the ionization interface, where after the effluent was introduced via an electrospray ionization (EPI) interface (Sciex, Toronto, ON) into a 4000 Q TRAP mass spectrometer. For maximal sensitivity and for linearity of the response, the mass spectrometer was operated in multiple-reaction monitoring (MRM) mode at unit mass resolution. Peaks were identified by comparison of retention time and mass spectra of standards. For each component, one ion transition was monitored, sirolimus: 931.6 → 864.4 (collision energy: 23 V), and tacrolimus: 821.5 → 768.4 (collision energy: 26 V). The following MS parameters were used: curtain gas: 10 psi; ion spay voltage: 5500 V; source temperature: 360 °C; gas flow 1:50 psi; gas flow 2:40 psi; decluster potential: 80 V and entrance potential: 10 V. Data were analyzed with Analyst software version 1.6.2 (Applied Biosystems, Nieuwerkerk a/d IJssel, The Netherlands). Tacrolimus peak areas were corrected for the sirolimus recovery, and concentrations were calculated using a tacrolimus reference line ranging from 0.5 ng to 1000 ng/mL which was linear in this range (r = 0.999).

    Synovial fluid analysis

    The synovial fluid WBC and TP concentrations were determined using a Coulter Counter® Z1 (Beckman Coulter, Inc.) and refractometer, respectively [36,37], following clinical practice. Synovial fluid samples were also evaluated for glycosaminoglycan (GAG) concentrations, as a marker for proteoglycan release, using a modified 1,9-dimethylmethylene blue dye-binding assay as previously described [38]. To check for possible damage to the collagen network of the cartilage, we also measured the concentration of C2C, a neo-epitope present on collagenase-cleavage fragments of type II collagen, employing a commercial ELISA kit (IBEX Technologies, Quebec, Canada) in accordance with the manufacturer’s recommendations. All assays have previously been validated for use in the horse [38,39].

    The prostaglandin E2 (PGE2) synovial fluid concentrations were determined by high-performance liquid chromatography (HPLC)tandem mass spectrometry (MS/MS) analysis as described previously [40]. Briefly, samples were recovered in a total volume of 1 ml of 15% (v/v) methanol + 0.5% glacial acetic acid in the presence of 10 pg/µL 16,16-dimethyl PGF2α that served as an internal standard. Samples were separated on a C18 SPE columns. The eicosanoids were eluted with 2 × 0.35 ml ethyl acetate and evaporated to dryness under nitrogen. Evaporated samples were reconstituted in 50 μl of 50% ethanol and subject to HPLC–MS analysis. Multiple reaction monitoring (MRM) was used as described previously [40]. The supernatant was transferred into an autosampler vial, and a 10 μl sample was injected onto a Luna C18 (2.5 μm 100 × 3 mm; Phenomenex, Torrance, CA, USA). Separation was performed at a flow rate of 200 μl/min with a total run time of 25 min. The mobile phases consisted of 0.02% glacial acetic acid in water (A) and 0.02% glacial acetic acid in acetonitrile (B). The following gradient was applied to the column; A/B vol/vol: 0–1 min, 80/20; 1–17 min, 63/37 to 52/48; 17–18 min, 52/48 to 13/87; 18–23 min, 0/100 and 24-25 min, 80/20 at ambient temperature. The effluent was introduced to an EPI interface into a 4000 QTRAP mass spectrometer. For maximal sensitivity and for linearity of the response, the mass spectrometer was operated in multiple-reaction monitoring (MRM) mode at unit mass resolution. Peaks were identified by comparison of retention time and mass spectra of standards. For each component, one ion transition was monitored, PGE2: 351.2 → 271.2 (collision energy: -25 V), and 16,16-dimethyl PGF (IS): 381.2 → 319.2 (collision energy: -35 V). The following MS parameters were used: curtain gas: 10 psi; ion spay voltage: -4,500; source temperature: 350 °C; gas flow 1:50 psi; gas flow 2:40 psi; decluster potential: -100 V and entrance potential: -10 V. Data were analyzed with Analyst software version 1.6.2 (Applied Biosystems, Nieuwerkerk a/d IJssel, The Netherlands). PGE2 peak areas were corrected for the IS recovery, and concentrations were calculated using a PGE2 reference line ranging from 10 to 1000 pg on column which was linear in this range (r = 0.99).

    PGE2 results from 8 hours had to be excluded because of too many missing samples for that timepoint, due to insufficient amount of synovial fluid. Furthermore, at 1, 3 and 5 weeks C2C synovial fluid levels were not determined because of limited synovial fluid volume availability.

  5. High resolution figures will be included in which the indicators for the p-values are clearer. If it is still not clear enough we will increase the size, however 4 symbols in a larger font do not fit in the width of the bars, therefore, we feel the current size is the best option.

  6.  We have adjusted this throughout the manuscript. 

We hope that after these alterations the manuscript meets your expectations and that you find it suitable for publication. 

Best regards, 

Lotte Groen

Reviewer 3 Report

This paper reports interesting and important results in the field of veterinary medicine. I suggest its publication after some revisions indicated below. The most important drawback of this manuscript is the lack of monospheres characterisation. The authors are suggested to provide the detailed characterisation of size distribution, loading efficiency, chemical characterisation, etc. Even if they followed any previous study on the same monospheres, this should be provided in current submission, too. Only once these data are provided, a careful evaluation of this paper can be performed.

Some technical issues: the authors are suggested to provide a more detailed description on histology/microscopy, also it appear that magnification might be wrong in Fig 5 caption, also the scale bars in Fig 5 are not visible well.

Author Response

Dear reviewer, 

Thank you for evaluating our manuscript.

We have added the table below with the characterisation that was performed after section 2.1.

Table 1: Overview of monospheres with their characteristics

Unloaded

Tacrolimus-loaded

Polymer

16[PDLA-PEG1000]-84[PLLA]

16[PDLA-PEG1000]-84[PLLA]

Average particle size

37 µm

39 µm

Morphology

Smooth and non-porous1

Smooth and non-porous1

FK506 loading

N.A.

9.2%

Encapsulation efficiency

N.A.

92%

Injection volume per joint

3 mL

3 mL

Monospheres injected per joint

200 mg

200 mg

Dose FK506 injected per joint

0 mg

18.4 mg

1Comparable as shown previously [32].

Considering section 2.6 (histology/microscopy): Thank you for your thoroughness, the magnification is 10x indeed. We have increased the thickness of the scalebars. Furthermore, we added information to section 2.6 to make it more complete.

2.6. Histological analysis and grading of articular cartilage and synovial lining

Six weeks after the start of the study, the animals were euthanized. Directly after, macroscopic scoring of the injected joints was performed according to McIlwraith et al. [41]. This system scores gross changes of the cartilage surface. More specifically, wear lines, erosions and palmar arthrosis were scored 0 - 3, where 0 = no changes, 1 = 1 - 2 partial thickness lines or < 5 mm diameter erosions, 2 = 2 - 5 partial or 1-2 full thickness lines, or > 5 mm diameter erosions, and 3 = > 5 partial or > 2 full thickness lines, or full thickness erosions. Next, samples were harvested and processed for histology as follows; in the middle carpal joints, opposing articular weightbearing surfaces (i.e. third and radial carpal bone articular surface) were harvested, in the talocrural joints the articular surface of the medial talar ridge was harvested. These osteochondral explants were decalcified in 0.5M EDTA. Furthermore, for each joint approximately 10 mm2 of synovial membrane was harvested randomly throughout the joint. All samples were fixed in buffered formaldehyde 4% solution (Klinipath, Duiven, The Netherlands) and embedded in paraffin. Sections of 5 µm thick were deparaffinized, and cartilage samples were stained with Safranin O/Fast Green to stain glycosaminoglycans red and collagen green, while synovial membrane samples were stained with eosin and cell nuclei were counterstained with heamatoxylin, as previously described by Gawlitta et al. [42]. All sections were then mounted in DPX (Millipore, USA), and micrographs were taken with an optical microscope (Olympus BX51, Olympus, Germany). A modified Mankin scoring system for histopathological grading was performed on osteochondral samples and a microscopic grading system on synovial membrane sections as described in McIlwraith et al., table V and VI respectively [41]. All scorings were performed by two independent researchers (SC and SP) in a blinded fashion.

We hope the manuscript meets your expectations after these alterations and that you find it suitable for publication. 

Best regards, 

Lotte Groen

Reviewer 4 Report

Review report

The present study entitled: “Sustained intra-articular release and biocompatibility of tacro-2 limus (FK506) loaded monospheres composed of [PDLA-PEG]-3 b-PLLA multi-block copolymers in healthy horse joints” investigated the safety, joint-biocompatibility of FK-506-loaded monospheres using an equine model. The manuscript is relatively well performed and of interest. Some important points and suggestions are listed below and needed to be well-elucidated before it could be considered for publication.

  1. The administration of therapeutic molecules through the intraarticular route has already been well-established for OA management for its clear benefit over systemic treatment, such as lower side effect, reduced cost, decreased systemic exposure and increased local concentration. In this study, FK-506 monosphere was injected intraarticularly and its effects on joint homeostasis was studied. This is an interesting study but many questions needed to be extensively addressed.

  1. The qualitative and quantitative composition of synovial fluid would change after monosphere injection, characterized by an increase in volume and fluctuation in viscosity. Marked reduction/change in viscosity is associated with a hyaluronic acid degradation which lead to joint degradation. Therefore, we believe that the physiochemical properties (e.g. rheological property, viscosity) of FK-506 monosphere should be well studied and demonstrated.

  1. For normal joint, the amount of synovial fluid is negligible and can be hardly aspirated. Could the authors describe the method of synovial fluid collection from horse joint throughout the study period? Do the authors perform joint Lavage?
  2. The authors claimed that they have performed macroscopic scoring of the injected joint (line 206-207). This part of data (macroscopic data, gross appearance, scoring, etc.) is missing. Please provide. It would be quite persuasive if the data is present.
  3. Horse joint sample is large. Please elucidate how the authors should determine which part of the joint specimens were analyzed histologically.
  4. For the in vivo study: All four joints of the same animal have been injected with different materials, therefore, strictly speaking there would be no true control existed in this study. Shouldn’t the lameness score be compared with normal healthy horses? I would say that the lameness score would be hardly compared by this study design.
  5. Any changes in body weight of horse noted throughout the experiment?
  6. Normal saline should be harmless and biologically tolerable while injected into joints. A transient increase of PGE2 was believed to be attributed to puncture injury to the joint. Could the authors explain why high level of PGE2 was detected in horse joint receiving NS injection?
  7. WBC should not be used as an isolated biomarker for joint inflammation, I don’t think WBC is sensitive enough for detecting subtle joint destruction. Some other inflammatory markers such as IL-6, MMP-3, MMP-13 are of paramount importance and more sensitive indicators of the cartilage degrative process.

Author Response

Dear reviewer, 

Thank you for evaluating our manuscript.

  1. Thank you for your compliment.
  2. Indeed the injection of a fluid in the articulating joint can have an influence on the viscosity of synovial fluid. The effect of 200mg monospheres in 3 ml saline is, however, neglectable when compared to the effect of 3 ml saline alone on the viscosity of synovial fluid, as the monospheres take up less than 7% of the total injected volume. For microspheres it is not possible to measure viscosity and rheological property as they are not liquid but solid. The other properties of both unloaded and tacrolimus-loaded monospheres were added in a new table after section 2.1.

    Table 1: Overview of monospheres with their characteristics

    Unloaded

    Tacrolimus-loaded

    Polymer

    16[PDLA-PEG1000]-84[PLLA]

    16[PDLA-PEG1000]-84[PLLA]

    Average particle size

    37 µm

    39 µm

    Morphology

    Smooth and non-porous1

    Smooth and non-porous1

    FK506 loading

    N.A.

    9.2%

    Encapsulation efficiency

    N.A.

    92%

    Injection volume per joint

    3 mL

    3 mL

    Monospheres injected per joint

    200 mg

    200 mg

    Dose FK506 injected per joint

    0 mg

    18.4 mg

    1Comparable as shown previously [32].

  3. At time point 0 (before injecting anything) it was possible to aspirate 1 – 2 ml of SF from the middle carpal and talocrural joints. These joints had been selected for that purpose, as they are easy to inject a substance and/or aspirate SF. Throughout the study we could retrieve limited amounts of SF without lavage. As lavage would influence the results of the study, we chose not to perform this. The horse is because of this easiness of retrieving SF an excellent model for the purpose of research of the articulating joint.

  4. Unfortunately no macroscopic pictures were made, as all joints looked perfect without any wear lines or other imperfections. This was confirmed by the histological data. The scores were noted straight after harvesting, which we added to section 2.6. If you think this is necessary, we could add a table as supplemental material, however, we think a table with only zeros is not of added value for the manuscript.

  5. We have added the following sentence to section 2.6: “In the middle carpal joints, opposing articular weightbearing surfaces (i.e. third and radial carpal bone articular surface) were harvested. In the talocrural joints the articular surface of the medial talar ridge was harvested.Furthermore, for each joint approximately 10 mm2 of synovial membrane was harvested randomly throughout the joint.”

  6. The lameness scores are compared with the individual scores pre-injection, therefore the baseline scores serve as control.

  7. We only weighed the horses at the beginning of the study. Based on visual inspection, the horses did not change in body condition in any relevant way.

  8. Normal saline is indeed biologically tolerable, but it does provoke changes in the joint. Previous studies on the influence of arthrocentesis itself (Van den Boom et al. Equine Vet. Journal, 2005) and on repeated intra-articular saline administration (Cokelaere et al., Eur. J Pharm Biopharm, 2018) have shown that saline provokes a mild transient inflammatory response. As PGE2 is a very sensitive biomarker for synovial inflammation, a temporary increase is also seen after saline intra-articular injections. Furthermore, saline is a substance with a different composition than synovial fluid, which, although only temporary, changes joint homeostasis with a transient increase of PGE2.  

  9. We agree that more parameters would have given a more complete indication of possible joint damage and, if possible, it would certainly be of added value to include more inflammatory markers in future studies. Unfortunately, as mentioned in section 4.2.2, limited amounts of synovial fluid obliged us to make a choice in biomarkers. WBC influx, total protein and PGE2 give a more general indication of the general inflammatory response and combined with the clinical symptoms, macroscopic and microscopic evaluation, it does give a good indication of biological tolerance.

We hope the manuscript meets your expectations after these alterations and that you find it suitable for publication. 

Best regards, 

Lotte Groen

Round 2

Reviewer 3 Report

acceptable as is

Reviewer 4 Report

The reviewer would like to congratulate the authors and expresses the significance and novelty of this manuscript. I think the authors have already addressed all the raised concerns and points proposed by the reviewer.

The reviewer feel that the manuscript is now suitable for publication and looks forward to better feedback from readers by the scientific community.